# Chemical Recycling of Used Printed Circuit Board Scraps: Recovery and Utilization of Organic Products

**Se-Ra Shin** [†] [iD], **Van Dung Mai** [†] [iD] **and Dai-Soo Lee \*** [iD]

Department of Semiconductor and Chemical Engineering, Chonbuk National University, 567 Baekje-daero, Deokjin-gu, Jeonju 54896, Korea; srshin89@jbnu.ac.kr (S.-R.S.); dungmv1983@gmail.com (V.D.M.)
**\*** Correspondence: daisoolee@jbnu.ac.kr; Tel.: +82-63-270-2310
† These authors contributed equally to this work.

**Abstract:** The disposal of end-of-life printed circuit boards (PCBs) comprising cross-linked brominated epoxy resins, glass fiber, and metals has attracted considerable attention from the environmental aspect. In this study, valuable resources, especially organic material, were recovered by the effective chemical recycling of PCBs. Pulverized PCB was depolymerized by glycolysis using polyethylene glycol (PEG 200) with a molecular weight of 200 g/mol under basic conditions. The cross-linked epoxy resins were effectively decomposed into a low-molecular species by glycolysis with PEG 200, followed by the effective separation of the metals and glass fibers from organic materials. The organic material was modified into recycled polyol with an appropriate viscosity and a hydroxyl value for rigid polyurethane foams (RPUFs) by the Mannich reaction and the addition polymerization of propylene oxide. RPUFs prepared using the recycled polyol exhibited superior thermal and mechanical properties as well as thermal insulation properties compared to conventional RPUFs, indicating that the recycled polyol obtained from the used PCBs can be valuable as RPUF raw materials for heat insulation.

**Keywords:** chemical recycling; glycolysis; used printed circuit board; recycled polyol; rigid polyurethane foam

---

## 1. Introduction

The development of various electronic and electrical equipment continuously generates a large amount of electronic and electrical equipment wastes [1–6]. Printed circuit boards (PCBs) constitute one of the essential components in electronic and electrical products. PCBs contain cross-linked brominated epoxy resins, metals, and glass fibers, which are difficult to decompose and reuse due to the inherent insolubility of the epoxy resins comprising a cross-linked network structure [1,7–11]. Thus, the appropriate disposal of used PCBs (UPCBs) has become an ongoing important issue from the environmental aspect [2,3,12].

UPCBs can be disposed by various methods, including landfills, incineration, pyrolysis, and chemical recycling [5]. Landfills and incineration can cause several environmental problems due to the presence of heavy metals and hazardous components in UPCBs [7]. On the other hand, thermal and chemical recycling is efficient for disposing UPCBs because of the separation and recovery of valuable materials from UPCBs. Several studies have reported the recovery of valuable resources from UPCBs by various methods [3,4,7,13–16]. Veti et al. reported the recycling of UPCBs to recover valuable metals [16] and found that lead (Pb), tin (Sn), and copper (Cu) are efficiently recovered by magnetic and electrostatic separation, followed by additional recovery via the electrowinning process. Zhu et al. reported a new technology for the recovery of valuable materials from UPCBs [13] by using 1-ethyl-3-methylimidazolium tetrafluoroborate [EMIM+] [BF$_4$$^-$] at 260 °C and found that solders, Cu,

glass fibers, and brominated epoxy resins are completely separated from UPCBs. Quan et al. examined the pyrolysis of UPCBs and successfully obtained the separated pyrolysis products comprising metals, glass fibers, and pyrolysis oil from UPCBs [3]. The pyrolysis oil derived from brominated epoxy resins contains a high concentration of phenol and phenol derivatives. Thus, pyrolysis oil is reused as a phenolic derivative to prepare a new phenol–formaldehyde resin. In addition, the authors expected that the glass fibers obtained from the pyrolysis residue can be applied as reinforcement filler for sheet molding compound and bulk molding compound.

The pyrolysis of UPCBs is one of the efficient recycling methods for the rapid recovery of valuable metals and glass fibers [1,3,9,14,15,17]. However, pyrolysis generally requires high temperature (>300 °C) and generates toxic organic materials [2–4,7]. In particular, hazardous gases including bromine compounds are released [10,18,19]. Nevertheless, owing to its high efficiency and short residence time, pyrolysis has been frequently employed for the recycling of UPCB to obtain valuable materials [18,20]. Chemical recycling, including glycolysis, aminolysis, and alcoholysis, demonstrates promise for the depolymerization of thermoplastic and thermosetting resins [11,21–25]. Typically, the chemical recycling of polymers involves the chain scission into small molecules by using hydroxyl or amine groups of solvolytic agents via transesterification [24]. Therefore, UPCBs mostly comprising brominated epoxy resins as the organic material are depolymerized, and the metallic and inorganic materials (i.e., metals and glass fiber) can be separated from the organic material (mostly brominated epoxy resin) during the process. Besides, organic products comprising hydroxyl-terminated groups and abundant aromatic rings can be converted to cost-effective raw chemicals to prepare a new polymer, especially polyurethane, and inorganics can be reused as reinforcements or fillers [12,13,21,26,27]. However, there are few reports on the utilization of the organic product from UPCBs for polyurethanes [21,28,29].

In this study, valuable resources, especially organic material were recovered from UPCBs via glycolysis. The pulverized UPCBs were depolymerized using polyethylene glycol (PEG) with a molecular weight of 200 g/mol under basic conditions. To optimize the depolymerization of UPCBs, the UPCB to PEG 200 ratio, reaction temperature, and reaction time were varied. Nonmetallic components such as organic material (depolymerization product) and glass fibers were collected and characterized. The organic product recovered after glycolysis contains phenolic and aliphatic hydroxyl end groups; thus, it can be used as a polyol to prepare rigid polyurethane foams (RPUFs). However, a solid organic product was obtained, and the hydroxyl value was not appropriate to prepare RPUFs. Thus, the organic product is liquefied and modified into recycled polyol for RPUFs via the Mannich reaction and the addition polymerization of propylene oxide. The recycled polyol obtained after the modification of the organic product was incorporated for manufacturing RPUFs, replacing 60 wt% of conventional polypropylene glycol (PPG) for RPUF. Besides, the RPUFs prepared using the recycled polyol exhibited improved thermal and mechanical properties compared to conventional RPUF. A flow chart for the chemical recycling of PCBs is given in Figure 1. To the best of our knowledge, this is the first report on the utilization of the organic product from UPCBs to prepare a recycled polyol for rigid polyurethane foams.

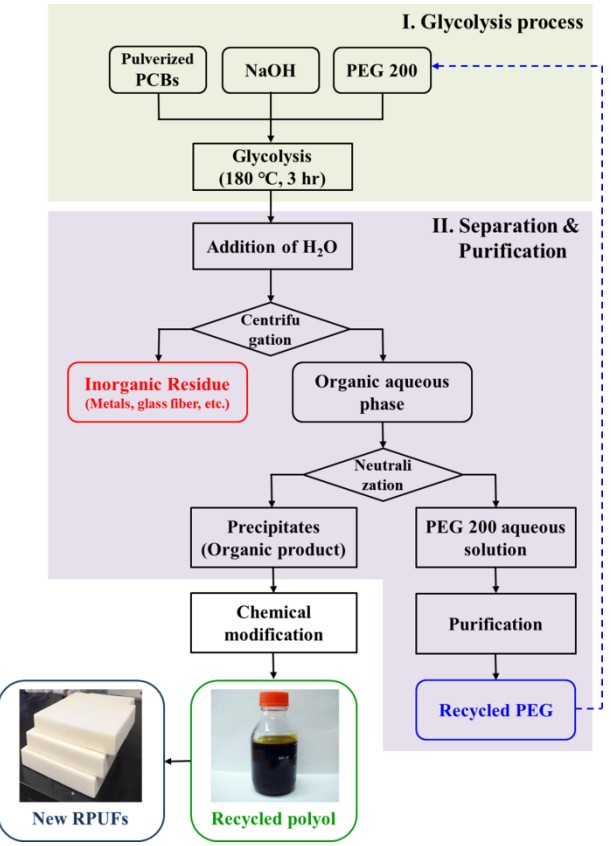

**Figure 1.** Flow chart for chemical recycling of pulverized printed circuit boards (PCBs).

## 2. Materials and Methods

### 2.1. Materials

UPCBs were collected from end-of-life computers and pulverized to remove metallic components by the density difference. PEG with a molecular weight of 200 g/mol (PEG 200) was purchased from Aldrich (Yong-In, Korea). Sodium hydroxide (NaOH) and potassium hydroxide (KOH), as well as a 0.5 M hydrochloric acid solution, were purchased from Daejung Chemicals & Metals (Si-Heung, Korea). Diethanolamine (Aldrich), a 37% formaldehyde solution (Aldrich), and propylene oxide (PO) (SKC chemicals, Ul-San, Korea) were used to modify the pre-polyol. A conventional PPG based on sugar/glycerin with a hydroxyl number of 450 mg KOH/g (JOP-0585) was supplied by Jungwoo Fine Chem Co., Ltd. (Ik-San, Korea). Polymeric 4,4′-diphenylmethane diisocyanate (pMDI, Cosmate M200) was purchased from Kumho Mitsui Chemicals (Yeo-Su, Korea). Dimethylcyclohexylamine (Polycat® 8, PC-8) from Air Products and Chemicals (Allentown, PA, USA) was used as amine catalysts. A silicone surfactant (B-8462) and tris(1-chloro-2-propyl) phosphate (TCPP) were purchased from Evonik (Essen, Germany) and Aldrich, respectively. As blowing agents, SOLKANE® 365/227 from Solvay (Brussels, Belgium) and distilled water were used. JOP-0585 was dried at 80 °C for 24 h under vacuum prior to use. All chemicals were used as received.

### 2.2. Glycolysis of Pulverized UPCB

The glycolysis of pulverized PCBs after the removal of metallic components was carried out under basic conditions. NaOH was dissolved in PEG 200 (NaOH:PEG 200 = 1:11 by mole) at 80 °C in a four-neck round-bottom flask, equipped with a reflux condenser and mechanical stirrer, under nitrogen, followed by the addition of a determined amount of PCB powder. The reaction temperature was allowed to increase to 180 °C. After 3 h, the system was cooled to room temperature, and distilled

water was added to the mixture to decrease its viscosity. The inorganic residue, mostly glass fibers, was separated from the aqueous solution by centrifugation. The washing of the residue by water and separation were repeated several times until the pH of the aqueous solution reached 7. The aqueous solution containing the glycolysis product and residual PEG 200 was neutralized by the addition of 0.5 M HCl, followed by the precipitation of the glycolysis product from the aqueous phase. The precipitates were collected, washed with water, and dried in a convection oven at 100 °C to remove water. After complete drying, a solid glycolysis product was obtained (referred to as pre-polyol). To optimize the efficiency of depolymerization: (i) the PCB to PEG 200 mass ratio was changed by 1/6, 1/7, and 1/9; (ii) glycolysis temperature was varied to 160, 180, and 200 °C; and (iii) glycolysis time was varied by 1, 2, 3, 4, and 5 h. The yield of the glycolysis product obtained from each experiment was estimated by thermogravimetric analysis.

### 2.3. Preparation of the Recycled Polyol

The solid pre-polyol was liquefied by the Mannich reaction and addition polymerization of PO, affording recycled polyol for RPUFs. The pre-polyol was obtained by glycolysis at 180 °C for 3 h, with a hydroxyl value of 226.9 mg KOH/g (Table 1). Briefly, the Mannich reaction and alkoxylation of pre-polyol were carried out as follows. A predetermined amount of pre-polyol (1.0 mol), diethanolamine (2.2 mol), and water were added to a round-bottom reactor equipped with a mechanical stirrer, thermometer, and nitrogen inlet. The reaction temperature was increased to 90 °C, and a formaldehyde solution (2.2 mol) was slowly added into the flask for the Mannich reaction. After 3 h at 120 °C, dehydration was carried out under reduced pressure until the water content decreased to less than 0.1%. Subsequently, the obtained Mannich adduct was modified by addition polymerization with PO, affording recycled polyol with a suitable hydroxyl value for RPUFs in autoclave. A dark brown liquid as the final product, recycled polyol, was collected and dehydrated at 80 °C under reduced pressure. The hydroxyl value of the product was measured by titration following ASTM D 4704. Table 1 summarizes the characteristics of conventional PPG (JOP-0585), pre-polyol, and recycled polyol. Fourier-transform infrared spectrum (FTIR) presented the following (Figure S1, cm$^{-1}$): 3376 (−OH), 2966 ($sp^3$ C−H), 2932 ($sp^3$ C−H), 2874 ($sp^3$ C−H), 1658 (Aromatic ring), 1509 (Aromatic ring), 1458, 1374, 1302, 1119, 1080, 934, 876, 840, and 584.

**Table 1.** Characteristics of JOP-0585, pre-polyol, and recycled polyol.

| | JOP-0585 | Pre-Polyol | Recycled Polyol |
|---|---|---|---|
| Viscosity (Pa·s) | 5.0 | - | 2.7 |
| Hydroxyl value (mg KOH/g) | 450.0 | 226.9 | 460.0 |
| Acid value (mg KOH/g) | >0.1 | >1.0 | 1.2 |
| Color | Light yellow | Dark brown | Dark brown |
| Br% | 0 | 15.0 ± 1.0 [a] 14.12 ± 1.04 [b] | 1.80 ± 0.13 [b] |

[a] Br% of pre-polyol was determined by EDS analysis. [b] Br% was determined by relative ratio of peak area% from pyrolysis gas chromatography–mass spectroscopy–electron capture detection (Py–GC/MS/ECD).

### 2.4. Preparation of RPUFs

RPUFs with different contents of recycled polyol were prepared in two steps. First, predetermined amounts of the surfactant, catalyst, phosphate, and blowing agents were added to the polyol and homogeneously mixed under mechanical stirring. Second, a precalculated amount of pMDI was added to the polyol mixture and vigorously mixed under mechanical stirring at 6000 rpm for 7 s. The mixture was rapidly poured into a steel mold (300 mm × 300 mm × 50 mm) with a lid. After curing at 60 °C for 20 min, the RPUFs were demolded and stored for at least 24 h at room temperature before the characterization. The NCO index was maintained at a constant value, 120. The recycled polyol contents with respect to the total polyol weight were increased from 0% to 60%. For comparison, the RPUF

prepared using only conventional PPG was designated as CON. Table 2 summarizes the formulations of the RPUFs with different contents of the recycled polyol.

**Table 2.** Sample code and formulation for rigid polyurethane foams (RPUFs).

| Sample Code | CON | P20 | P40 | P60 |
|---|---|---|---|---|
| | **(Composition by wt%)** | | | |
| *Polyol part* | | | | |
| JOP-0585 | 100.0 | 80.0 | 60.0 | 40.0 |
| Recycled polyol | - | 20.0 | 40.0 | 60.0 |
| B-8462 | 2.0 | 2.0 | 2.0 | 2.0 |
| PC-8 | 3.0 | 3.0 | 2.0 | 1.5 |
| TCPP | 15.0 | 15.0 | 15.0 | 15.0 |
| Water | 1.5 | 1.5 | 1.5 | 1.5 |
| 365/227 | 35.0 | 35.0 | 35.0 | 35.0 |
| *Isocyanate part* | | | | |
| NCO Index | 120 | 120 | 120 | 120 |

*2.5. Characterization*

The chemical structures of the pre-polyol and the recycled polyol were examined by [1]H and [13]C NMR spectroscopy (600 MHz, JNM-ECA600, JEOL Ltd., Tokyo, Japan) in DMSO-$d_6$ at room temperature. Fourier transform infrared (FTIR) spectra (FTIR 2000, JASCO, Easton, MD, USA) were recorded in the wavenumber range from 4000 to 500 cm$^{-1}$ at a resolution of 4 cm$^{-1}$. Scanning electron microscopy (SEM) images (JSM 6400, JEOL Ltd., Akishima, Tokyo, Japan) were recorded to examine the morphology of the glass fiber after depolymerization at an accelerating voltage of 20 kV. The content of bromine in the pre-polyol and recycled polyol was determined by energy-dispersive spectroscopy (EDS) (JSM-6400, JEOL Ltd., Akishima, Tokyo, Japan), pyrolysis gas chromatography–mass spectroscopy (Py–GC/MS) (QP2010 plus, Shimadzu, Kyoto, Japan), and gas chromatography–electron capture detection (GC/ECD) (6890N, Agilent Technologies, Santa Clara, CA, USA). EDS analysis was carried out with gold-coated pre-polyol powder at an accelerating voltage of 20 kV. Py–GC/MS with medium polar column (DB–624, 30 m × 0.251 mm × 1.40 mm, Agilent Technologies) was employed for the identification of the overall compounds in the pre-polyol and the recycled polyol. The pyrolysis temperature was 670 °C and the oven was held at 40 °C for 3 min, and then ramped to 260 °C at 10 °C/min. The brominated phenolic compounds thereof were identified by GC/ECD with medium polar column (DB–624, 30 m × 0.251 mm × 1.40 mm, Agilent Technologies). The analysis conditions of GC/ECD were identical to GC. The content of metals in recycled polyol such as Al, Ni, Cu, Cd, and Pb was determined by inductively coupled plasma–mass spectrometry (ICP/MS).

To characterize the reactivity of RPUFs, the characteristic times, including the cream time, gel time, and tack-free time, were estimated. The compressive strength of the RPUFs was estimated by using a universal testing machine according to ASTM D 1621. Cubic samples with a size of 40 mm × 40 mm × 40 mm were prepared, and the blowing direction and direction perpendicular to blowing were evaluated. The compressive strength of five specimens per sample was estimated, and the average values were calculated. To eliminate the effect of density, the measured compressive strengths were normalized for the blowing direction and direction perpendicular to blowing as follows (Equation (1)):

$$\sigma_n = \sigma(40/\rho)^2 \left[ \{1 + \sqrt{(40/\rho_s)}\}/\{1 + \sqrt{(\rho/\rho_s)}\} \right]^2 \tag{1}$$

where $\sigma_n$ represents the normalized compressive strength; $\sigma$ represents the measured compressive strength; $\rho$ represents the density of RPUF (kg/m$^3$); and $\rho_s$ represents the density of the solid polyurethane matrix, which is given as 1200 kg/m$^3$ [30].

Thermal conductivities of the RPUFs were evaluated using a heat flow meter (HFM 436 Lambda, Netzsch, Selb, Germany) with two plane plates maintained at different temperatures (ASTM C 518).

Thermal conductivities of three specimens per sample were estimated, and the average values were obtained. The closed-cell content of the RPUFs with dimensions of 25 mm × 25 mm × 25 mm was estimated on an ULTRAPYC 1200e (Quantachrome, Boynton Beach, FL, USA) pycnometer according to ASTM D 6226. The closed-cell contents of five specimens per sample were estimated and averaged. The cell morphology of the RPUFs was examined by SEM (JSM-6400, JEOL Ltd., Akishima, Tokyo, Japan) at an accelerating voltage of 20 kV. The samples were coated with gold to avoid the charging of the electrons. Thermogravimetric analysis (TGA) was carried out on a Q50 system from TA Instruments (New Castle, DE, USA). Approximately 10 mg of sample placed on a platinum pan was heated from ambient temperature to 800 °C at a heating rate of 20 °C/min under nitrogen atmosphere. TGA measurements were carried out three times per sample and representative data were used for analysis. Dynamic mechanical property measurements of the RPUFs were carried out on a dynamic mechanical analyzer (DMA, Q800, New Castle, DE, USA) from TA Instruments in the tension mode from 30 °C to 250 °C at a heating rate of 3 °C/min (a frequency of 1 Hz and an amplitude of 15%).

## 3. Results and Discussion

### 3.1. Glycolysis of UPCBs

PEG is not only the solvent but also the co-catalyst for the glycolysis of pulverized PCBs. A sufficient amount of PEG is required to disperse PCB powders and combine with sodium hydroxide to promote glycolysis. To identify the effect of the PCB content on the efficiency of glycolysis, the PCB to PEG mass ratio was varied from 1:6 to 1:9 (Table 3). Figure 2 shows the effect of the PEG content on the weight loss of the glass fiber under the same glycolysis conditions (at 180 °C after 3 h). With the decrease of PEG content in the decomposition of PCB from 86.1 wt% to 84.4 wt%, the weight loss of glass fiber at 800 °C increased from 4% to 6%, respectively, and the yield increased from 89.74% to 92.47% (Table 3). However, with the increase in the PCB content from 86.1 wt% to 88.5 wt%, the weight loss of the glass fiber after glycolysis was almost the same.

**Table 3.** Composition and yield of the decomposition of used PCBs (UPCB) with polyethylene glycol (PEG).

| Codes | Before Decomposition | | | | | | | | After Decomposition | | | | | | Yield |
| | PCB | | PEG | | NaOH | | Glass Fiber | | Decomposed Product | | Recycled PEG | | | |
| | g | wt% | g | wt% | g | wt% | g | wt% | g | wt% | g | wt% | % |
| 1:6 | 20.01 | 14.08 | 120.01 | 84.42 | 2.14 | 1.51 | 13.79 | 9.91 | 6.21 | 4.46 | 119.19 | 85.63 | 89.74 |
| 1:7 | 20.04 | 12.33 | 140.03 | 86.14 | 2.49 | 1.53 | 13.61 | 8.54 | 6.43 | 4.04 | 139.26 | 87.42 | 92.08 |
| 1:9 | 25.12 | 9.91 | 224.31 | 88.50 | 4.02 | 1.58 | 16.98 | 6.84 | 8.13 | 3.28 | 223.05 | 89.88 | 92.47 |

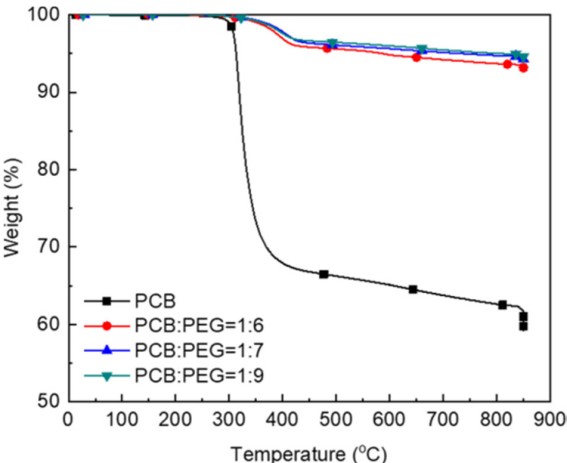

**Figure 2.** Thermogravimetric analysis (TGA) curves of PCB and the glycolysis products at different ratios of PCB to polyethylene glycol (PEG 200).

Figure 3 shows the effect of temperature on the glycolysis of pulverized PCBs. The weight loss values for the glass fiber at 800 °C after glycolysis at 180 °C and 200 °C for a reaction time of 3 h were almost the same (~4.50%, Figure 3a). However, with the decrease in the glycolysis temperature to 160 °C, the weight loss at 800 °C considerably increased (29.28%). Based on the weight loss at 800 °C, the glycolysis yield was estimated. Figure 3b shows the effect of the glycolysis temperature on the glycolysis yield. With the increase in the glycolysis temperature from 160 °C to 180 °C, the glycolysis yield sharply increased. Moreover, similar glycolysis yields were observed at 180 °C and 200 °C. Hence, the optimal reaction temperature was determined to be 180 °C. The effect of temperature can be explained in terms of thermodynamics. At low temperature, the thermal energy is not sufficient for breaking chemical bonds, thereby affording a low yield.

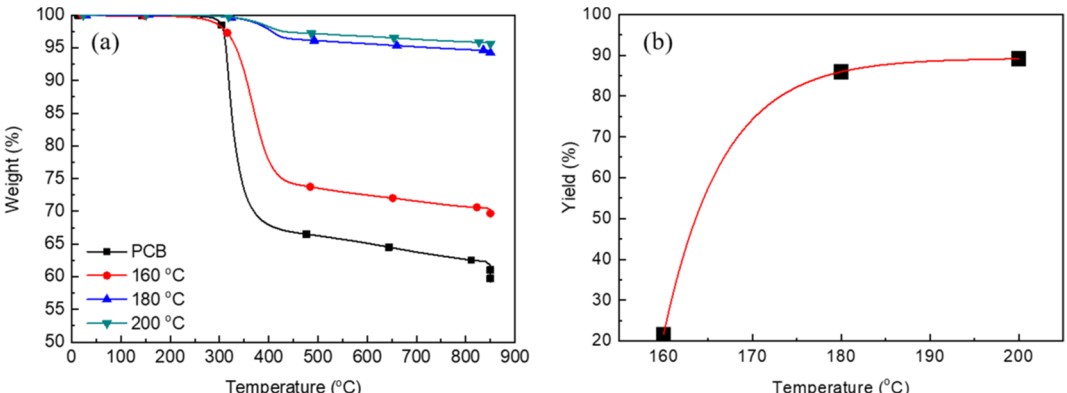

**Figure 3.** TGA analyses of PCB and glass fibers after the reaction at difference reaction temperatures (**a**) and the glycolysis yield for difference reaction temperature (**b**).

Figure 4 shows the effect of the glycolysis time on the efficiency of glycolysis at 180 °C. The weight loss was clearly ~5% after 2 h at 180 °C (Figure 4a). After 3 h, the weight loss did not decrease further because the decomposition reaction reached equilibrium. Figure 4b shows the effect of the reaction time on the glycolysis yield. The glycolysis yield sharply increased with the reaction time until 2 h at 180 °C, following which the yield did not significantly vary with the further increase in the reaction time.

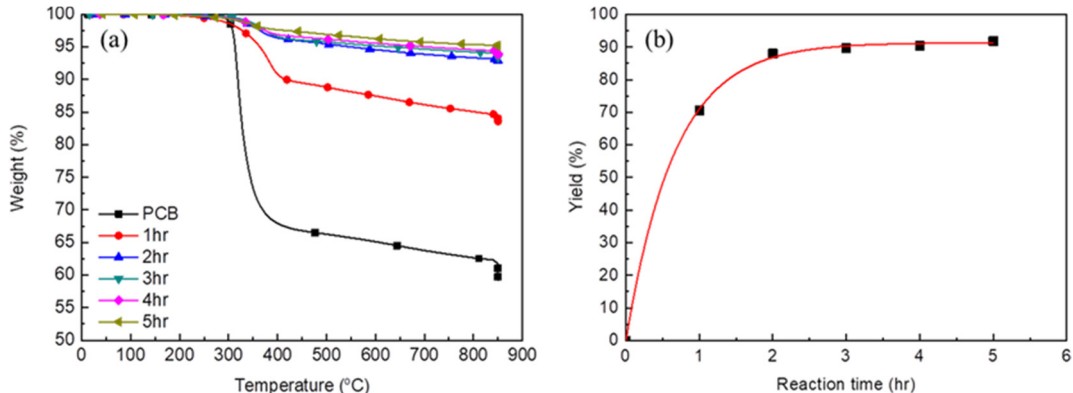

**Figure 4.** TGA analyses of PCB and glass fiber after the reaction for difference reaction time (**a**) and the glycolysis yield for difference reaction time (**b**).

SEM images also revealed the effects of the reaction time on the glycolysis efficiency (Figure 5). After the decomposition for 1 h, the glass fibers retained a considerable amount of polymer on their surfaces, and the glass fibers were still stuck. After 2 h, the glass fibers were separated, but few polymer particles were still retained on the glass fiber surface. Hence, the brominated epoxy resin was not completely decomposed, and a 100% yield was not achieved. After 3 h of glycolysis, the surface of the

glass fibers appeared smooth and clean. Therefore, it was concluded that the optimum reaction time for glycolysis is 3 h. The glass fibers obtained after the glycolysis can be used as reinforcing fillers for polymer composites as many previous researchers reported on the reutilization of glass fibers recycled from PCBs [21,27,31,32]. Zheng et al. studied the reutilization of nonmetals recycled from UPCBs as reinforcing fillers in polypropylene composites [27]. They demonstrated that the incorporation of nonmetals recycled from PCBs significantly improved the tensile and flexural properties of the polypropylene composites. Sun et al. investigated sound absorption performance of glass fibers recycled from UPCBs [32]. They found that the recycled glass fibers showed excellent absorption ability in broad-band frequency range, implying the promising candidates for sound absorption materials. In this study, we concentrated on the utilization of organic products from UPCB.

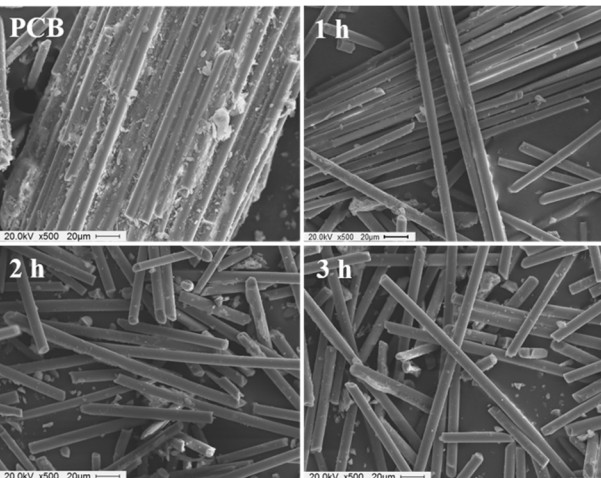

**Figure 5.** Scanning electron microscopy (SEM) images of the PCB powder and glass fiber after the reaction for 1 h, 2 h, and 3 h.

PCBs are comprised of reinforcing glass fibers, metals, and epoxy resins containing brominated flame retardants. Bisphenol A and diglycidyl ether of bisphenol A epoxy resin cured using an anhydride hardener has been widely used for manufacture of PCBs. To limit the possible hazards associated with scarce fire resistance and the inherent flammability of these materials, flame retardants are typically added in their formulation. Typical flame retardants include tetrabromobisphenol A, which is substituted for bisphenol A in the epoxy resin. After the reaction, the epoxy resin was decomposed and converted to monomers in the final product. Figure 6 shows the FTIR spectrum of the final glycolysis product: A broad peak corresponding to the O–H stretching vibrations was observed at 3296 $cm^{-1}$. Sharp absorption peaks were observed between 2871 and 2965 $cm^{-1}$, which were assigned to C–H stretching vibrations. A peak observed at 1705 $cm^{-1}$ corresponds to the stretching vibrations of C=O in the ester bonds of the curing agent. Absorption peaks between 1475 and 1609 $cm^{-1}$ were attributed to the stretching vibration of the aromatic rings. Peaks were observed at 1246 $cm^{-1}$ and 1101 $cm^{-1}$ corresponding to C–O and O–C stretching vibrations, respectively. The peak observed at 832 $cm^{-1}$ corresponds to H–$C_{arom}$ bending vibrations.

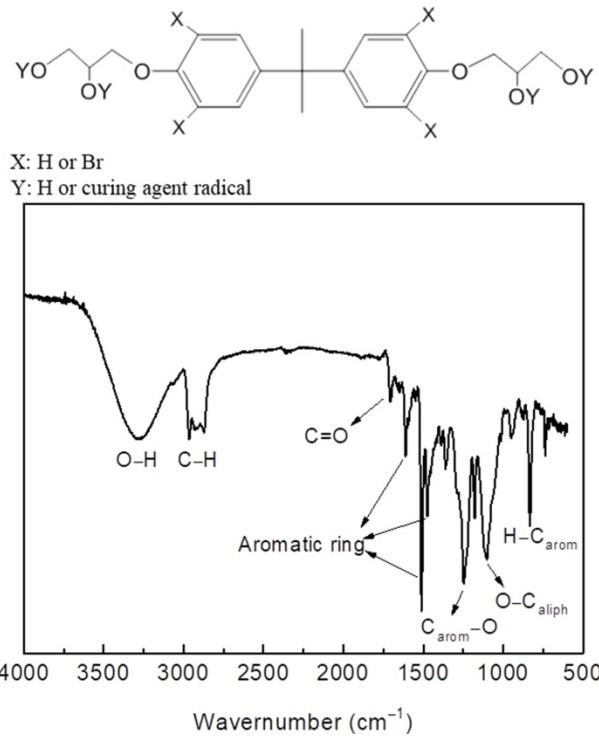

**Figure 6.** Fourier-transform infrared spectrum (FTIR) of the glycolysis product (pre-polyol).

In addition, the [1]H NMR and [13]C NMR spectra were recorded to examine the chemical structures of the final product as shown in Figure 7. In the [1]H NMR (Figure 7a), peaks observed between 6.6 and 7.3 ppm correspond to the aromatic ring protons. The signals ranging from 3.3 to 4 ppm correspond to the aliphatic chain protons. The peak at 1.6 ppm is derived from the methyl group proton of the bis-phenol A. Figure 7b shows the [13]C NMR of the final product. Peaks observed between 112 and 155 ppm correspond to the aromatic ring carbons. Peaks observed at 60–72 ppm are associated with the aliphatic carbons of bisphenol A diglycidyl ether. Peaks observed at 30 ppm and 43 ppm are identical to the methyl group and methane carbons, respectively, verifying the presence of bisphenol A diglycidyl ether and tetrabromobisphenol A, while the curing agent structures in the final product are hardly identified.

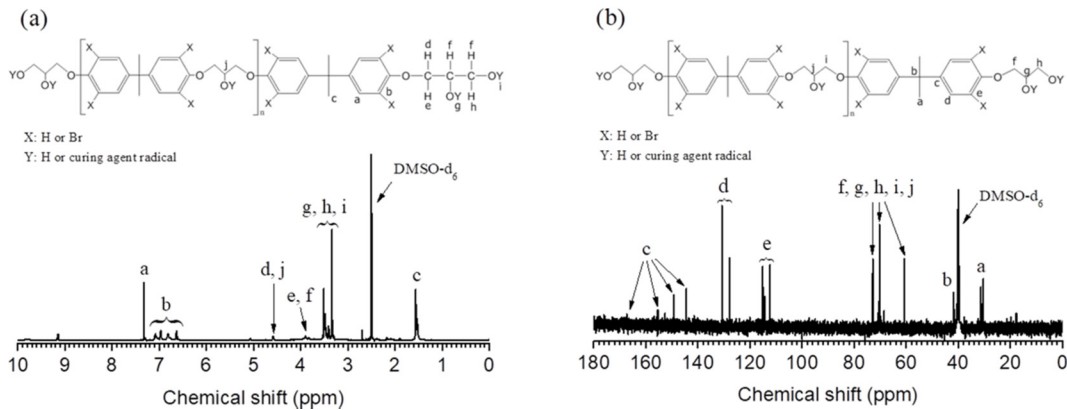

**Figure 7.** [1]H NMR (**a**); and [13]C NMR (**b**) spectra of the glycolyses product (pre-polyol) in DMSO-$d_6$.

A model of the epoxy resin based on bisphenol A diglycidyl ether and tetrahydrophthalic anhydride was successfully decomposed using a PEG 200/NaOH system. Scheme 1 shows the plausible reaction mechanism for glycolysis. The ester bonds of the cured epoxy resin are broken by

hydrolysis, affording an organic sodium salt. The organic salt structure includes bisphenol A diglycidyl ether, tetrabromobisphenol A, and the curing agent, which is soluble in a mixture of PEG and water. After the addition of HCl, the sodium cations in organic salts are replaced by protons, and the organic salts are converted to polyol ethers, which are not soluble in water because of their high molecular weight and branched structure; hence, the polyol ethers precipitate to form a solid phase.

**Scheme 1.** Plausible reaction mechanism for the glycolysis.

PEG collected after the first recycling step by using virgin PEG 200 was continuously used for the decomposition of PCBs under the optimal reaction conditions of a PCB to PEG mass ratio of 1:7, a reaction temperature of 180 °C, and a reaction time of 3 h. Recycled PEG was purified by the removal of the sodium chloride formed during the reaction between the sodium organic salt and HCl. The addition of tetrahydrofuran (THF) led to the removal of sodium chloride from PEG. During the addition of THF, sodium chloride precipitated to a mixture of recycled PEG due to the different solubilities of PEG and sodium chloride in THF. The precipitates were separated using a filter, and the filtrate contained PEG and THF, followed by the extraction of the recycled PEG using a rotary evaporator based on the different boiling points of PEG and THF. This process was repeated until sodium chloride was completely removed. Finally, PEG was further dried overnight in a vacuum oven at 60 °C to completely remove THF. In addition, PEG after the second and third recycling steps was collected by the above process after reusing the PEG obtained from the first and second recycling steps, respectively. Figure 8a shows the TGA curves of the glass fiber collected after the reaction with virgin PEG 200, as well as the first, second, and third recycled PEG. The weight loss of the glass fiber after glycolysis of PCBs with the PEG recycled from the first step slightly decreased in comparison to virgin PEG 200, while those of the glass fiber after the glycolysis of PCBs with the second and third recycle PEG were considerably reduced. The increased weight loss implied the reduction of the glycolysis yield (Figure 8b). With increasing number of cycles for recycling PEG, the glycolysis yield decreased because of the degradation of PEG 200 and formation of a lower-molecular-weight PEG during glycolysis. Lower- or higher-molecular-weight PEG compared to PEG 200 did not coordinate with sodium ions to form a catalyst for glycolysis.

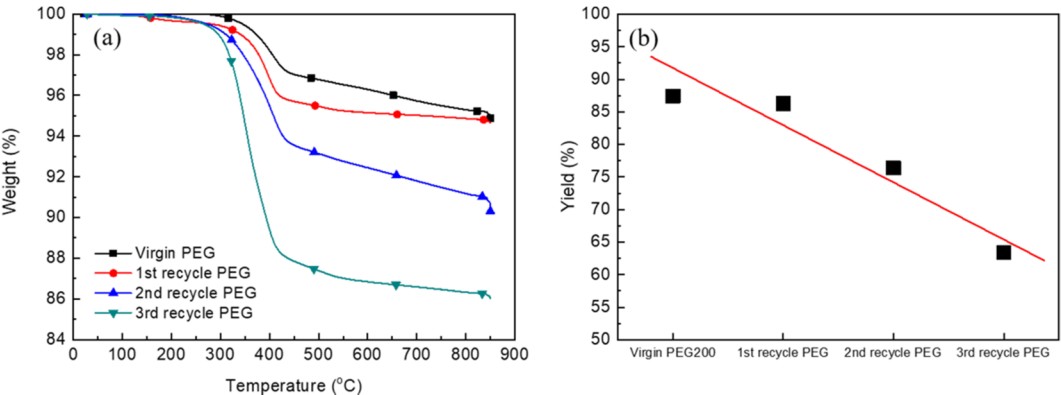

**Figure 8.** TGA curves of the glass fiber after the decomposition with virgin PEG 200 and recycled PEG (**a**) and the glycolysis yield of the recycled PEG (**b**).

### 3.2. Characteristics of RPUFs Prepared from the Recycled Polyol Based on the Glycolysis Product of UPCBs

The glycolysis product and recycled polyol were composed of phenol and phenolic derivatives, and brominated compounds as confirmed by Py−GC/MS (Figures S2 and S3). The representative chemical structures detected in Py−GC/MS are summarized in Tables S1–S3. As shown in Tables S1 and S2, the glycolysis product showed similar chemical structure to conventional brominated epoxy resin. It demonstrates that the organic parts of UPCBs are comprised of diglycidyl ether of bisphenol A containing brominated bisphenol A as flame retardant. The representative brominated compounds in glycolysis products and recycled polyol identified from Py−GC/MS/ECD are summarized in Tables S5 and S6. The bromine content determined by EDS and GC/ECD was estimated as the relative ratio by examining the conventional brominated epoxy resin with already known bromine content. Table 1 summarizes the bromine content of the pre-polyol and recycled polyol. The bromine content of pre-polyol determined by EDS and GC/ECD was similar, i.e., 15.0% and 14.2%, respectively. The recycled polyol was a viscous liquid; thus, the bromine content was only measured by GC/ECD. The bromine content of the recycled polyol was 1.8%, indicating a considerably lower value compared to pre-polyol, probably related to the detachment of the bromine atoms from the benzene ring during the chemical modification. Furthermore, the increased molecular weight after the modification of pre-polyol led to the dilution of the bromine atom concentration of the overall molecules. Thus, the brominated phenolic derivatives were hardly detected in Py−GC/MS/ECD spectrum of recycled polyol (Table S2). Table 4 summarizes the content of metals in recycled polyol as determined by ICP/MS. The content of the representative elements, such as Al, Ni, Cu, Cd, and Pb, were evaluated. As Al and Cu were the major components of PCB; their concentrations were greater than those of various metal elements. On the other hand, concentrations of heavy metals, such as Ni, Cd, and Pb, were almost not detected (~0). Notably, the recycled polyol obtained after chemical recycling and modification can replace the petrochemical polyols for manufacturing new RPUFs without environmental hazards. Furthermore, the recycled polyol containing abundant phenol derivatives is thought to contribute to the enhanced physical properties of the resulting foams.

**Table 4.** Content of metals in the recycled polyol.

| Element | Concentration (%) |
|---------|-------------------|
| Al | 0.0266 |
| Ni | 0.0000 |
| Cu | 0.0066 |
| Cd | 0.0000 |
| Pb | 0.0010 |

Table 5 summarizes the reactivities of recycled polyol during foaming. All of the characteristic times (i.e., cream time, gel time, and tack-free time) were more rapid with the increase in the recycled polyol content despite the low content of the amine catalyst (PC-8). The accelerated reactivities were related to the catalytic effect of the tertiary amine in the recycled polyol formed during the Mannich reaction.

**Table 5.** Characteristic times and density of RPUFs.

| Sample | CON | P20 | P40 | P60 |
|---|---|---|---|---|
| Cream time (s) | 19 | 15 | 14 | 13 |
| Gel time (s) | 108 | 70 | 55 | 48 |
| Tack-free time (s) | 170 | 120 | 95 | 85 |
| Density (kg/m$^3$) | 44.1 | 44.0 | 44.7 | 45.0 |

Figure 9 shows the cross-sectional SEM images of RPUFs with different contents of recycled polyol. All RPUFs exhibited a polyhedral and uniform closed-cell morphology without shrinkage or collapse even at a high recycled polyol content (60 wt%). With the increase in the recycled polyol content, the average cell size clearly decreased considerably. The rapid reactivity of the recycled polyol should restrict the expansion and coalescence of bubbles during foaming, leading to a smaller cell size of the resulting foams. Recycled polyol is the PO adduct with an abundant benzene ring; hence, it may exhibit a high affinity toward the raw materials for RPUFs, especially pMDI. High compatibility among the components contributed to the formation of a finer, homogenous cell structure.

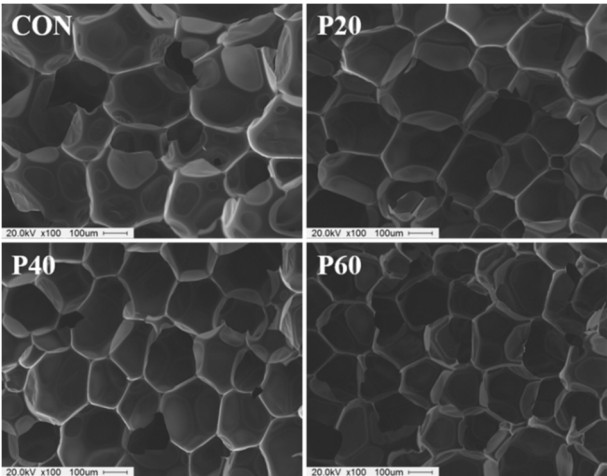

**Figure 9.** SEM micrographs of the RPUFs prepared from recycled polyol.

In addition, the closed-cell content of RPUF is one of the important parameters for thermal insulating materials [33,34]. The thermal conductivities of carbon dioxide (0.0150 W/m°C) and HFA-365/227 (0.0107 W/m°C) are considerably less than that of air (0.0240 W/m°C). Thus, blowing gas entrapped in a closed cell cannot escape from the cell, maintaining a low thermal conductivity. Figure 10 shows the closed-cell content of RPUFs prepared with different contents of recycled polyol. The closed-cell content increased to the maximum of 93.2% for P40 from 89.1% for CON. Although the closed-cell content for RPUF containing 60 wt% of the recycled polyol was slightly decreased, those of all of the RPUFs prepared using recycled polyol were greater than that of CON, revealing that the incorporation of recycled polyol for preparing RPUFs does not cause issues in the cell structure, including opening and breaking.

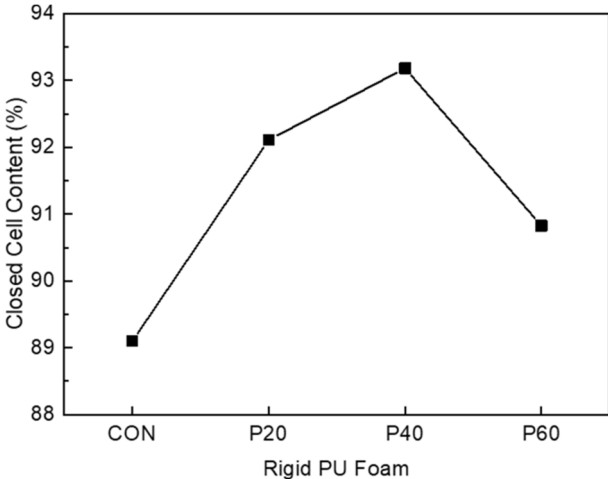

**Figure 10.** Closed-cell content of the RPUFs prepared from recycled polyol.

Figure 11 shows the thermal conductivities of RPUFs with different contents of recycled polyol. Typically, the thermal conductivity of RPUFs is strongly affected by the cell size, closed-cell content, thermal conductivity of the blowing gas entrapped in the cells, and density [30,35–37]. In this study, RPUFs prepared using recycled polyol exhibited a smaller cell size and higher closed-cell content compared to CON. From these contributions, the thermal conductivity of recycled-polyol-based RPUF is less than that of CON. In particular, P40 exhibited the minimum thermal conductivity of 0.0184 kcal/mh°C, which was in good agreement with the highest closed-cell content.

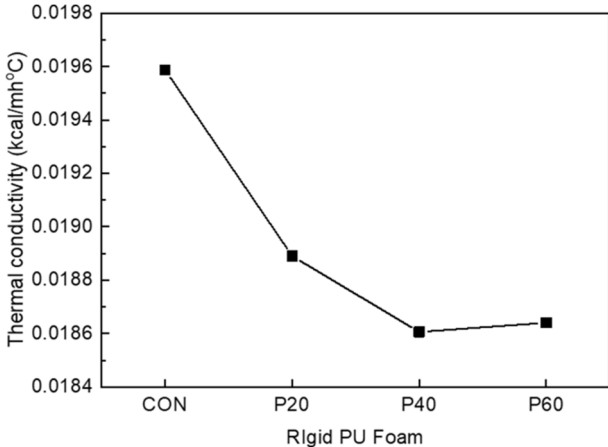

**Figure 11.** Thermal conductivity (K-factor) of RPUFs with different contents of recycled polyol.

Figure 12 shows the normalized compressive strength in the direction of blowing (B) and in the direction perpendicular to blowing (P) at a 10% strain of RPUFs with different contents of recycled polyol. Recycled-polyol-based RPUFs exhibited superior compressive strength in both the measured directions compared to CON due to the smaller cell size [38]. Besides, a significant amount of the aromatic ring in the recycled polyol would render additional strength to the cell, leading to the improved compressive strength of RPUFs. The maximum compressive strengths were observed in both directions for the RPUF containing 40 wt% of the recycled polyol. The compressive strengths of P40 in the direction of blowing and in the direction perpendicular to blowing were improved by 62% and 18%, respectively, compared to CON. However, with the further increase in the recycled polyol content of up to 60 wt%, the compressive strength decreases, possibly related to the presence of slight defects in the cell structure; this decreased strength was already confirmed by the decrease in

the closed-cell content of P60. However, its compressive strengths (B: 0.128 MPa and P: 0.169 MPa) were comparable to CON (B: 0.112 MPa and P: 0.178 MPa).

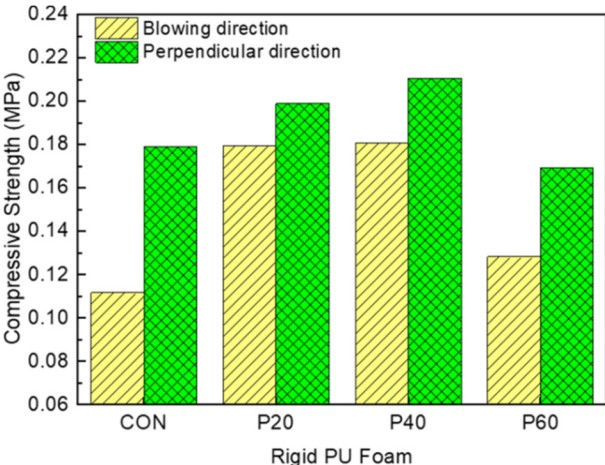

**Figure 12.** Compressive strengths measured in the blowing direction (B) and direction perpendicular to the blowing (P) of RPUFs with different contents of recycled polyol.

The thermal degradation behavior of RPUFs with different contents of recycled polyol was evaluated by TGA under nitrogen. Thermogravimetry (TG) and derivative TG (DTG) thermograms of RPUFs are shown in Figure 13. The TGA curve of CON exhibited a typical thermal degradation behavior for RPUF. Three major thermal decomposition steps were observed. The first step corresponded to the thermal degradation of the urethane unit at 190–220 °C; the second step corresponded to the thermal degradation of polyol at 300–360 °C; and the third step observed at 480–520 °C corresponded to the thermal degradation of aromatic isocyanate and hydrocarbons [39–41]. On the other hand, recycled-polyol-based RPUFs exhibited additional thermal degradation at 429 °C, corresponding to the degradation of recycled polyol. Thus, the weight loss at this temperature increases with the recycled polyol content. $T_{max3}$ corresponding to the thermal decomposition of aromatic groups shifted to lower temperature, eventually overlapping possibly with the thermal degradation of recycled polyol. Table 6 summarizes the maximum degradation temperature ($T_{max}$) at each degradation stage and residue% after the degradation of RPUFs. The thermal degradation of the hard segment occurred at lower temperature with increasing content of recycled polyol. However, in the major thermal degradation stage at which the most weight loss ($T_{max2}$) occurred, the RPUF with more recycled polyol was degraded at the higher temperature. In particular, the $T_{max2}$ of the polyol increased by 25 °C from 314.0 °C for CON to 339.2 °C for P60. In addition, RPUFs based on the recycled polyol retained a high amount of residual char after heating in TGA. Clearly, the incorporation of recycled polyol can enhance the thermal stability due to the presence of thermally stable bromine atoms and the aromatic ring [42–46].

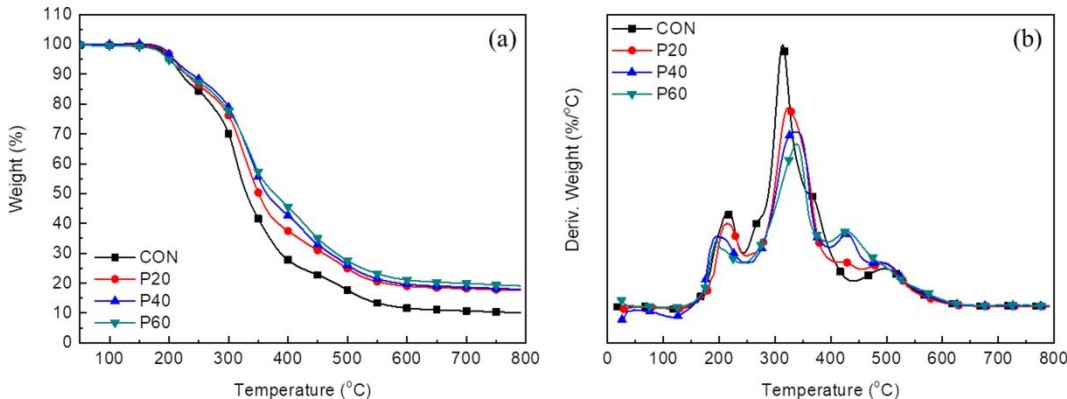

**Figure 13.** Thermogravimetry (TG) (**a**); and derivative TG (DTG) (**b**) thermograms of RPUFs with different content of recycled polyol under nitrogen atmosphere.

**Table 6.** Maximum degradation temperature ($T_{max}$), percentage of residue after degradation in TGA, and $T_g$ of RPUFs with different contents of recycled polyol.

| Sample | $T_{max1}$ (°C) | $T_{max2}$ (°C) | $T_{max3}$ (°C) | Residue (%) | $T_g$ (°C) |
|--------|-----------------|-----------------|-----------------|-------------|------------|
| CON | 216.8 | 314.0 | 496.6 | 10.1 | 134.6 |
| P20 | 211.7 | 324.5 | 494.6 | 17.6 | 149.4 |
| P40 | 200.2 | 338.1 | 488.2 | 18.0 | 165.9 |
| P60 | 195.0 | 339.2 | - | 19.1 | 167.6 |

Figure 14 shows the results obtained by DMA (i.e., storage modulus (G′), loss modulus (G″), and tan delta) of RPUFs with different contents of the recycled polyol. Table 6 summarizes the glass transition temperature ($T_g$), which is determined by the maximum point of the loss modulus and $T_g$ values. The G′ (rubbery region) and $T_g$ values of RPUFs increased with the recycled polyol content due to the high amount of the aromatic ring in the recycled polyol. At a temperature greater than $T_g$ at which the molecular chain was softened to move, the presence of a rigid aromatic ring in the polymer network restricted the movement of the molecular chain, leading to increased G′ and $T_g$. With the increase in the recycled polyol content, the tan delta curves broadened, and the maximum point was hardly observed for the RPUFs containing greater than 40 wt% of the recycled polyol content (Figure 14c). Typically, the width of a tan delta curve is closely related to the phase separation between the soft and hard segments in a polyurethane network. The broadening of the tan delta curve revealed glass transition of the crosslinked polyurethanes [47–49]. As mentioned above, the recycled polyol used in this study is an aromatic ring-rich PO adducts; thus, it exhibits good compatibility with pMDI. It would induce the phase mixing of soft and hard segments in the polyurethane network, leading to an increase of loss modulus peak or tan delta peak temperatures for RPUF loaded with a high content of recycled polyol.

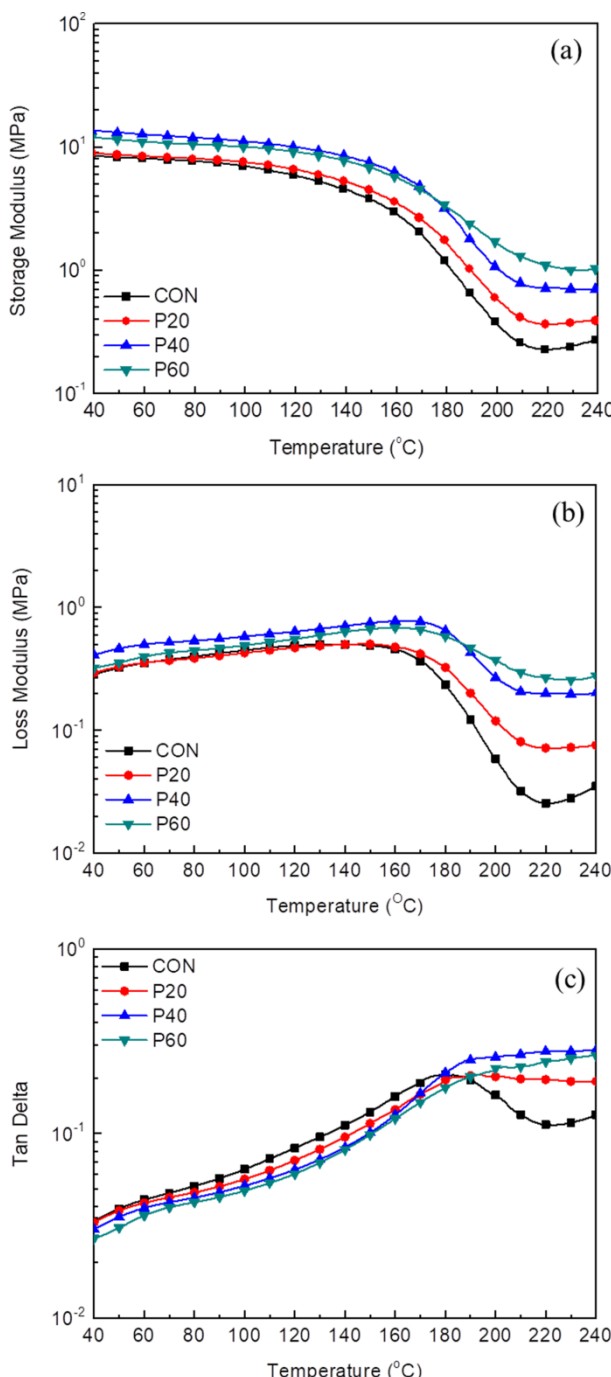

**Figure 14.** Dynamic mechanical analyzer (DMA) results of RPUFs with different content of recycled polyol as a function of temperature: storage modulus (G′) (**a**); loss modulus (G″) (**b**); and Tan delta (**c**).

## 4. Conclusions

In this study, pulverized UPCBs were successfully depolymerized by glycolysis with PEG 200 under basic conditions, and organic materials and glass fibers were recovered. In addition, the glycolysis conditions, including the PCB to PEG 200 mass ratio, reaction temperature, and reaction time, were optimized. The glass fibers obtained after glycolysis exhibited a smooth surface and short length without destruction, indicating that the glass fibers can be reused as reinforcements. Furthermore, the bromine content of the glycolysis products was ~15% with a significant amount of the aromatic groups, which was derived from the brominated epoxy resins of PCBs. Almost no heavy metals were detected in the organic components. PEG 200 recovered after glycolysis was collected

and reused again for the glycolysis of PCBs, and the glycolysis efficiency of recovered PEG 200 was obtained by the weight loss of the organic materials remaining in the glass fiber after glycolysis. The glycolysis yield slightly decreased with increasing the number of recycling PEG 200. The organic product (pre-polyol) obtained after glycolysis was converted to a recycled polyol via Mannich reaction and addition polymerization of PO to prepare RPUFs for thermal insulation. Besides, the recycled polyol could replace 60 wt% of conventional PPG for RPUFs without difficulties and deterioration of foam properties. The RPUFs based on recycled polyol exhibited superior compressive strength, thermal insulation property, and thermal stability to the conventional RPUF. The superior performance of RPUFs based on recycled polyol was attributed to the fine cell morphology as well as the presence of aromatic rings in the recycled polyol. It is inferred that the recycled polyol obtained from glycolysis and chemical modification of UPCBs can be promising feedstock for thermal insulating RPUFs.

**Supplementary Materials:** The following are available online at http://www.mdpi.com/2227-9717/7/1/22/s1, Figure S1: FTIR spectrum of recycled polyol obtained by modification of glycolysis product of UPCBs, Figure S2: Py–GC/MS of conventional brominated epoxy resin of tetrabromobisphenol A, Figure S3: Py–GC/MS of glycolysis product obtained from chemical recycling of UPCBs, Figure S4: Py–GC/MS of recycled polyol obtained from modification of glycolysis product, Table S1: Representative chemical structure of standard brominated epoxy resin from Py–GC/MS, Table S2: Representative chemical structure of glycolysis product by from Py–GC/MS, Table S3: Representative chemical structure of recycled polyol from Py–GC/MS, Table S4: Representative brominated compounds in conventional brominated epoxy resin by using Py–GC/MS/ECD, Table S5: Representative brominated compounds in glycolysis product by using Py–GC/MS/ECD, Table S6: Representative brominated compounds in recycled polyol by using Py–GC/MS/ECD.

**Author Contributions:** S.R.S., V.D.M., and D.S.L. contributed to conceptualization, methodology, design, and writing draft. S.R.S. and V.D.M. performed the investigation and data analysis.

**Funding:** This study was supported by the R & D Center for Valuable Recycling (Global-Top R & BD Program) of the Ministry of Environment (Project No: 2016002240004).

**Conflicts of Interest:** The authors declare no conflict of interest.

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
