# Peer review of "Chemical Recycling of Used Printed Circuit Board Scraps: Recovery and Utilization of Organic Products"

_processes, doi:10.3390/pr7010022_

Round 1
Reviewer 1 Report
Dear authors,
The management of wasted printed circuit boards represents an important issue that must be managed urgently. So, papers on this topic are always welcome.
The only point I must comment is about the overall quality of the English within the whole text.
A review of the language is needed before publication.
Best regards.
Author Response
Thank you for your review and comments on our manuscript.
Following your comments, the manuscript was revised.
Respond to Review 1 Comments
Comment: The only point I must comment is about the overall quality of the English within the whole text. A review of the language is needed before publication.
Respond: The manuscript has been carefully reviewed by an experienced editor whose first language is English and who specializes in editing papers written by scientists whose native language is not English.
Reviewer 2 Report
This paper presented study of recovering organic materials fromo WPCBs. The authors clearly illustrated the experiment design and result analysis. The paper is well structured. I believe the topic is of high importance and interest to public, especially in waste management. I recommend it publish in this journal.
My only suggestion is the literature review. The authors should summarize the research gap and indicate the significance of this study. In current introduction, it is not very clear.
Author Response
Thank you for your review and comments on our manuscript.
Following your comments, the manuscript was revised.
Respond to Review 2 Comments
Comment: My only suggestion is the literature review. The authors should summarize the research gap and indicate the significance of this study. In current introduction, it is not very clear.
Respond: In the revised manuscript, the significance of our work compared with the previous reports in the literatures was mentioned in highlighted lines 66~67 and 82~83.
Reviewer 3 Report
The potential reutilisation of glass fibres should be substantiated by some more figures or discussed more in detail.
Author Response
Thank you for your review and comments on our manuscript.
Following your comments, the manuscript was revised.
Respond to Review 3 Comments
Comment: The potential reutilisation of glass fibres should be substantiated by some more figures or discussed more in detail.
Respond: In the revised manuscript, previous reports in the literatures on the reutilization of glass fibers recycled from PCBs were described in the highlighted lines 243~254. In this study, we focused on the recovery and utilization of organic products obtained by chemical recycling of printed circuit boards (PCBs). Thus, we did not perform the further experiments for the reutilization of glass fibers.
Reviewer 4 Report
Manuscript with a title of “Chemical Recycling of Waste Printed Circuit Board Scraps: Recovery and Utilization of Organic Products” is presenting interesting experimental results on the recycling process of useful materials from used PCBs. The subject of the manuscript well stands in the scope of the Processes journal.
1. English writing is clear and correct, but my eyes catch a few errors that need editing. The author can fix these few errors during proofreading stage.
For example, using N-dash in numbers range (line 39, [3,4,7,13-16], which must be [3,4,7,13–16]). N-dash should also be used in page number range in the References section, for instance, 131–136, not 131-136 in Ref #1.
2. The Title is clear and sufficient. In the title, “Waste Printed Circuit Board Scraps” seems inaccurate. This research is based on the PCBs removed from used electronic devices, which are different from scarps and wastes from failed PCBs in the electronics manufacturing factories. Thus, I suggest changing it to ‘Used Printed Circuit Boards’. In this way the title will be: ‘Chemical Recycling of Used Printed Circuit Boards: Recovery and Utilization of Organic Products’.
3. The Abstract is sufficient and clear.
4. Keywords are fine and sufficient.
5. Introduction presents enough literature review and is clear.
6. The Materials and Methods section is clear.
7. In Results and Discussion section, the organization is fine.
With consideration of inhomogeneous composition of PCB, how 10 mg samples for TGA examinations could represent the real composition of PCB sample? Because of the inhomogeneity of PCB composition, it is expected to have significant errors in TGA examination. If there were replicates in the TGA examination, it should be explained, otherwise forget about it.
8. The Conclusions section is clear and sufficient.
9. Please check errors in the References and fix them all. For instance, in Ref #11, the journal is Polymer, not Poly. And, in Ref #14, “Waste Manag” should be ‘Waste Manage.’.
All journals’ names must be in abbreviated form, while in ref #44, it is the full name.
10. The Supplementary Materials document is clear and presents useful additional information about this research
The manuscript can be considered for publication in Processes after a minor revision of editing the writing and style errors.

Author Response
Thank you for your review and comments on our manuscript.
Following your comments, the manuscript was revised.
Respond to Review 4 Comments
Comment 1: English writing is clear and correct, but my eyes catch a few errors that need editing. The author can fix these few errors during proofreading stage. For example, using N-dash in numbers range (line 39, [3,4,7,13-16], which must be [3,4,7,13–16]). N-dash should also be used in page number range in the References section, for instance, 131–136, not 131-136 in Ref #1.
Respond 1: In the revised manuscript, N-dash (–) was used for the range of references instead of hyphen (-) following the comment.
Comment 2: The Title is clear and sufficient. In the title, “Waste Printed Circuit Board Scraps” seems inaccurate. This research is based on the PCBs removed from used electronic devices, which are different from scarps and wastes from failed PCBs in the electronics manufacturing factories. Thus, I suggest changing it to ‘Used Printed Circuit Boards’. In this way the title will be: ‘Chemical Recycling of Used Printed Circuit Boards: Recovery and Utilization of Organic Products’.
Respond 2: According to the comment, the title was changed to ‘Chemical Recycling of Used Printed Circuit Boards: Recovery and Utilization of Organic Products’ in the revision. ‘WPCBs (waste printed circuit boards)’ was also corrected to UPCBs or pulverized PCBs.
Comment 3: In Results and Discussion section, the organization is fine.
With consideration of inhomogeneous composition of PCB, how 10 mg samples for TGA examinations could represent the real composition of PCB sample? Because of the inhomogeneity of PCB composition, it is expected to have significant errors in TGA examination. If there were replicates in the TGA examination, it should be explained, otherwise forget about it.
Respond 3: PCB samples used in this study were pulverized into fine powder of round 100 meshes. Therefore, we could assume that PCB sample loaded on TGA was homogeneous. In the revised manuscript, it was explained that TGA measurements were carried out three times per sample and the representative data was used for analysis (line 190~191).
Comment 4: Please check errors in the References and fix them all. For instance, in Ref #11, the journal is Polymer, not Poly. And, in Ref #14, “Waste Manag” should be ‘Waste Manage.’. All journals’ names must be in abbreviated form, while in ref #44, it is the full name.
Respond 4: In the revision, all journals’ names were abbreviated in correct form and highlighted according to the comment.